# Deriving Six Components of Reynolds Stress Tensor from Single-ADCP Data

Sergey Bogdanov *, Roman Zdorovennov, Nikolay Palshin and Galina Zdorovennova *

Karelian Research Centre, Northern Water Problems Institute, Russian Academy of Sciences (NWPI), 185030 Petrozavodsk, Russia; romga74@gmail.com (R.Z.); npalshin@mail.ru (N.P.)
* Correspondence: sergey.r.bogdanov@mail.ru (S.B.); zdorovennova@gmail.com (G.Z.); Tel.: +7-9116660369 (G.Z.)

**Abstract:** Acoustic Doppler current profilers (ADCP) are widely used in geophysical studies for mean velocity profiling and calculation of energy dissipation rate. On the other hand, the estimation of turbulent stresses from ADCP data still remains challenging. With the four-beam version of the device, only two shear stresses are derivable; and even for the five-beam version (Janus+), the calculation of the full Reynolds stress tensor is problematic currently. The known attempts to overcome the problem are based on the "coupled ADCP" experimental setup and include some hard restrictions, not to mention the essential complexity of performing experiments. In this paper, a new method is presented which allows to derive the stresses from single-ADCP data. Its essence is that interbeam correlations are taken into account as producing the missing equations for stresses. This method is applicable only for the depth range, for which the distance between the beams is comparable to the scales, where the turbulence is locally isotropic and homogeneous. The validation of this method was carried out for convectively-mixed layer in a boreal ice-covered lake. The results of computations turned out to be physically sustainable in the sense that realizability conditions were basically fulfilled. The additional verification was carried out by comparing the results, obtained by the new method and "coupled ADCPs" one.

**Keywords:** full set of turbulent stresses; Acoustic Doppler current profilers; interbeam velocity correlations; ice-covered lakes; convectively-mixed layer; anisotropic turbulence

## 1. Introduction

Acoustic Doppler current profilers (ADCP) are currently viewed as one of the most powerful tools for geophysical flows studies. The product family of these devices includes a lot of versions, which differ from each other by the number of beams, transducer head design, carrier acoustic frequency, cell sizes and time measurements settings. This variety of the device parameters provides flexibility in deployment and makes it possible to adjust the measurements to the broad range of research needs in meteorological (e.g., [1]), oceanological (e.g., [2–4]) and limnological (e.g., [5,6]) studies.

In particular, under the requirement of flow horizontal homogeneity, ADCPs are widely used for mean velocity profiling, as was originally designed. The later instrumental development (e.g., "burst" time settings, velocity measurements extending to 'pulse coherent mode') makes it possible to achieve higher resolution and better accuracy of the measurements, thus triggering the use of ADCP for enhanced studies of turbulence parameters. Within this new domain of ADCP applications, meaningful results have been obtained in fine-scale studies, including the estimations of dissipation rates $\varepsilon$ [7–9].

At the same time some special methods have been developed for deriving the parameters of large-scale turbulence, with the special attention to the components $\left\langle u_i' u_j' \right\rangle$ of Reynolds stress tensor ($u_i'$ pulsation velocity components in orthogonal frame) (e.g., [3,4,10–12]). The equations for these components are derived from the directly available intensities $\left\langle b_i'^2 \right\rangle$

of the "beam" velocities pulsations $b_i'$. However, on the whole the problem remains challenging. Firstly, the number of ADCP beams usually varies from three to four (Janus configuration) and five (Janus+), and the system of equations is not complete. Secondly, in general case by calculating $\langle b_i'^2 \rangle$ one can obtain only the relationships between the different required components $\langle u_i' u_j' \rangle$, but not the explicit relations for each of them. For example, with three-beam ADCP, no explicit relations are available. As for the four-beam Janus configuration, after aligning the device axis with the mean velocity, by applying the so-called "variance method" one can derive two explicit relations for off-diagonal stresses (shear stresses), but that is all.

One of the ways to overcome the problem was presented in [13,14]. In both papers the main idea was based on ADCP coupling, when the experimental setup includes two rigidly connected ADCPs. In such a special setup, the design and implementation of the experiment become more complicated. Besides, in both cases the stresses derivation was conjugated with additional restrictions. The method suggested in [13] gave acceptable results only for the case when the axis of the second device was sufficiently (>20°) tilted to the vertical. However, this requirement is not recommended by the device manufacturers; besides, the tilting makes the horizontal homogeneity requirement tougher. In the method, presented in [14], the axes of both devices are vertical, and one pair of beams have the intersection point at some depth. However, in turn, this method also possesses restrictions: it is applicable only to a small range of depths, close to the depth of the intersection point.

In this paper, an alternative method for derivation of full Reynolds stresses from single-ADCP data is presented. As compared to the coupled-ADCP method, presented in [14], this new method is not restricted to stress computations for the special depth, and so gives the opportunity for stress profiling for a range of depths. The missing information is derived not from the additional beam data, but by taking into account the interbeam correlations of the velocity. It is worthy of note that usually these correlations are neglected, by suggesting the statistical independence of velocities at different beams. Meanwhile, in geophysical flows the integral scales of turbulence often are so large that the size of energy-containing eddies at some depths occurs commensurable with the distance between the beams. This is just the case, when beam velocities are correlated, and their covariance includes some "hidden" information, necessary for closing the equation system for $\langle u_i' u_j' \rangle$.

The new method of turbulent stress derivation was applied to the case of the convectively mixed layer (CML) which develops in lakes during under-ice inhomogeneous solar heating of the water column.

## 2. Method Description

*General Framework*

In what follows below the simplest ADCP version with three azimuthally symmetric beams is considered (Figure 1). For the standard configuration the angle $\alpha_0$ between the beam and vertical is 25°. As for the angle $2\alpha$ between any pair of beams, its value may be determined by the following expression, derived from pure geometrical analysis:

$$sin\alpha = \frac{\sqrt{3}}{2} sin\alpha_0$$

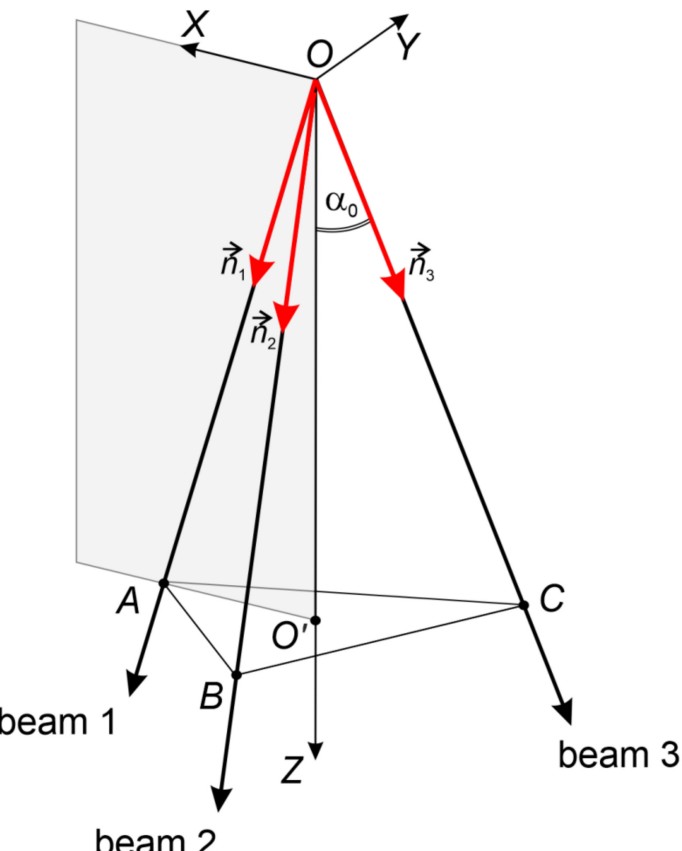

**Figure 1.** The coordinate system and main notations for three-beam ADCP configuration. Point O corresponds to the device head. Axis X lies in the plane AOO'.

In the orthogonal coordinate system *XYZ*, which is rigidly connected to the device, axis *X* is chosen in such a way that it lies on the same plane as beam 1 (Figure 1). In this frame of reference, the unit vectors $\overrightarrow{n}_1$, which identify the beams directions, have the following direction cosines:

$$\begin{cases} \overrightarrow{n}_1 = (sin\alpha_0, {}^\circ 0, {}^\circ cos\alpha_0), \\ \overrightarrow{n}_2 = \left(-\frac{1}{2}sin\alpha_0, {}^\circ - \frac{\sqrt{3}}{2}sin\alpha_0, {}^\circ cos\alpha_0\right), \\ \overrightarrow{n}_3 = \left(-\frac{1}{2}sin\alpha_0, {}^\circ \frac{\sqrt{3}}{2}sin\alpha_0, {}^\circ cos\alpha_0\right). \end{cases}$$

The trigonometric coefficients in the above expressions depend only on the basic angle $\alpha_0$. Later on the correspondent matrix is denoted by **N**.

Each of the beam velocities $\{b_i\}$, measured directly at points *A*, *B*, *C*, is connected to the orthogonal projections $u_1$, $u_2$, $u_3$ of the velocity $\overrightarrow{u}$ at the same points by the linear relations, e.g., $b_1(A) = \left(\overrightarrow{n}_1 \overrightarrow{u}(A)\right)$, or, equivalently (note the summation over the repeated indexes):

$$b_1(A) = N_{1j}u_j(A) \tag{1}$$

For the flows, which are homogeneous in the horizontal plane, $\langle u_i(A)\rangle = \langle u_i(B)\rangle = \langle u_i(C)\rangle \equiv \langle u_i\rangle$ so the mean values three equations of the type (1) take the form: $\langle b_i\rangle = N_{ij}\langle u_j\rangle$. As the result, the mean velocity components at the given depth are obtained directly as the convolution of the so-called transformation matrix $\hat{T} = \hat{A}^{-1}$ with the "vector" $(\langle b_1(A)\rangle, \langle b_2(B)\rangle, \langle b_3(C)\rangle)$ of mean beam velocities.

The same homogeneity assumption makes it possible to express the beam velocity pulsations intensities $\langle b_i'^2 \rangle$ through the turbulent stresses, e.g.,:

$$\langle b_1'^2 \rangle = \langle u_i'^2 \rangle sin^2\alpha_0 + \langle u_3'^2 \rangle cos^2\alpha_0 + 2\langle u_1'u_3' \rangle sin\alpha_0 cos\alpha_0 \tag{2}$$

Expression (2) and two similar ones (for beams 2 and 3) represent three linear equations for six target components $\langle u_i'u_j' \rangle$, so the system is incomplete. Moreover, the explicit relation for any stress through beam pulsation intensities is not available, as was mentioned in the introduction.

In the general case, the problem of yielding the turbulent stresses from three-beam ADCP data is highly problematic. For some flows, not only the external length scales, but also integral scales of turbulence are large enough as compared to the distance between beams. For such cases the correlations between beam velocities are not vanishing, and taking the values of $\langle b_i b_j \rangle$ into account gives the opportunity to overcome the problem of missing equations.

To implement this opportunity, the structural function (SF) of the general type $\widetilde{D}_{12}$ should be introduced into consideration. For beams 1 and 2, for example, this SF is defined as:

$$\widetilde{D}_{12} = \langle (b_1(A) - b_1(B))(b_2(A) - b_2(B)) \rangle \tag{3}$$

Under the assumption of local isotropy and homogeneity, the function $\widetilde{D}_{12}$ is presented through the longitudinal SF $D_{LL}$ (see Appendix A): $\widetilde{D}_{12} = \left( \frac{4}{3}cos^2\alpha - sin^2\alpha \right) D_{LL}$. So, after opening the brackets, and taking $\langle b_1(A)b_2(A) \rangle = \langle b_1(B)b_2(B) \rangle$ into account (horizontal homogeneity) Expression (3) takes the form:

$$\left( \frac{4}{3}cos^2\alpha - sin^2\alpha \right) D_{LL} = 2\langle b_1 b_2 \rangle - \, < b_1(A)b_2(B) > - \, < b_1(B)b_2(A) > \tag{4}$$

Here $\langle b_1 b_2 \rangle = \, < b_1(A)b_2(A) >$.

Longitudinal SF $D_{LL}$ is derived directly by calculating along-beam velocity correlations. The second term in the r.h.s. of Equation (4) is also available directly from experimental data. As for the last term in the r.h.s., after taking into account both the horizontal homogeneity and refection invariance, the following expression is obtained in [15]: $< b_1(A)b_2(B) > = \, < b_1(B)b_2(A) >$. So, all except $\langle b_1 b_2 \rangle$ terms in the Equation (4) are available from the experiment, and one may regard Equation (4) as the explicit expression for $\langle b_1 b_2 \rangle$. With regard to the presentation $\langle b_1 b_2 \rangle = \langle b_1'b_2' \rangle + \langle b_1 \rangle \langle b_2 \rangle$ this expression may be also written as:

$$\langle b_1'b_2' \rangle = \left( \frac{2}{3}cos^2\alpha - \frac{1}{2}sin^2\alpha \right) D_{LL} + \langle b_1(A)b_2(B) \rangle - \langle b_1 \rangle \langle b_2 \rangle \tag{5}$$

The similar presentations are valid for two remaining pairs of beams (13 and 23).

On the other hand, with presentation (1) in mind, three "beam stresses" $\langle b_i'b_j' \rangle (i \neq j)$ may be presented as the linear combination $N_{il}N_{jm}\langle u_l'u_m' \rangle$ of the Reynolds tensor components $\langle u_l'u_m' \rangle$ in the same way as Expression (2) for $\langle b_i'^2 \rangle$ were derived. For example:

$$\langle b_1'b_2' \rangle = sin^2\alpha_0(-\langle u_1'^2 \rangle/2 - \sqrt{3}\langle u_1'u_2' \rangle/2 + \langle u_1'u_3' \rangle(cot\alpha_0)/2 - \sqrt{3}\langle u_2'u_3' \rangle(cot\alpha_0)/2 + \langle u_3'^2 \rangle cot^2\alpha_0) \tag{6}$$

Expression (6) and two similar ones (for $\langle b_1'b_3' \rangle$ and $\langle b_2'b_3' \rangle$) one may regard as three missing equations for $\langle u_i'u_j' \rangle$. Together with Equation (2) (and two similar expressions —for $\langle b_2'^2 \rangle$ and $\langle b_3'^2 \rangle$) they form the closed system of linear inhomogeneous equations.

To represent this system in the compact form, it is reasonable to introduce into consideration the following "vectors":

$$B_i = \left( \left\langle b_1'^2 \right\rangle, \left\langle b_2'^2 \right\rangle, \left\langle b_3'^2 \right\rangle, \left\langle b_1' b_2' \right\rangle, \left\langle b_1' b_3' \right\rangle, \left\langle b_2' b_3' \right\rangle \right),$$

$$R_i = \left( \left\langle u_1'^2 \right\rangle, \left\langle u_1' u_2' \right\rangle, \left\langle u_1' u_3' \right\rangle, \left\langle u_2'^2 \right\rangle, \left\langle u_2' u_3' \right\rangle, \left\langle u_3'^2 \right\rangle \right).$$

With these notations the system of equations becomes:

$$B_i = M_{ij} R_j, \; i, j = 1 \dots 6 \tag{7}$$

The coefficient matrix **M** is derived directly from Equations (2) and (6) and similar ones:

$$\mathbf{M} = \sin^2 \alpha_0 \begin{pmatrix} 1 & 0 & 2\cot\alpha_0 & 0 & 0 & \cot^2\alpha_0 \\ 1/4 & \sqrt{3}/2 & -\cot\alpha_0 & 3/4 & -\sqrt{3}\cot\alpha_0 & \cot^2\alpha_0 \\ 1/4 & -\sqrt{3}/2 & -\cot\alpha_0 & 3/4 & \sqrt{3}\cot\alpha_0 & \cot^2\alpha_0 \\ -1/2 & -\sqrt{3}/2 & \cot\alpha_0/2 & 0 & -\sqrt{3}\cot\alpha_0/2 & \cot^2\alpha_0 \\ -1/2 & \sqrt{3}/2 & \cot\alpha_0/2 & 0 & \sqrt{3}\cot\alpha_0/2 & \cot^2\alpha_0 \\ 1/4 & 0 & -\cot\alpha_0 & -3/4 & 0 & \cot^2\alpha_0 \end{pmatrix}.$$

Summing up, it seems reasonable to stress some key points and the step by step procedure. The corresponding algorithm looks as follows:

1.  After proper choice of time averaging interval, the mean beam velocities $\langle b_i \rangle$, pulsation intensities $\langle b_i'^2 \rangle$ (Equation (2)) and correlations $\left\langle b_i' b_j' \right\rangle$ (Equation (5), $i \neq j$) are calculated directly from experimental data.
2.  For each beam the function $D_{LL}$ is calculated. After revealing the inertial interval, its extent is estimated, with the special attention to its upper scale limit $l$.
3.  The range of depths is chosen in such a way that the distance between beams does not exceed the scale $l$. The maximum depth $h$ is derived from inequality $AB < l$ (see Figure 1): $h < l / \left( \sqrt{3} \tan\alpha_0 \right)$.
4.  For chosen depths, the turbulent stresses are calculated directly by solving the system (7):

$$R_i = M^{-1}{}_{ij} B_j, \; i, j = 1 \dots 6 \tag{8}$$

Here the inverse matrix $\mathbf{M}^{-1}$ looks like (here $tan(\alpha_0)$ is shortly denoted as $t$):

$$\mathbf{M}^{-1} = \sin^{-2}\alpha_0 \begin{pmatrix} 4/9 & 1/9 & 1/9 & -4/9 & -4/9 & 2/9 \\ 0 & 1/\left(3\sqrt{3}\right) & -1/\left(3\sqrt{3}\right) & -2/\left(3\sqrt{3}\right) & 2/\left(3\sqrt{3}\right) & 0 \\ 2t/9 & -t/9 & -t/9 & t/9 & t/9 & -2t/9 \\ 0 & 1/3 & 1/3 & 0 & 0 & -2/3 \\ 0 & -t/\left(3\sqrt{3}\right) & t/\left(3\sqrt{3}\right) & -t/\left(3\sqrt{3}\right) & t/\left(3\sqrt{3}\right) & 0 \\ t^2/9 & t^2/9 & t^2/9 & 2t^2/9 & 2t^2/9 & 2t^2/9 \end{pmatrix}.$$

## 3. Experimental Setup and Results

Method validation was carried out with the velocity data, obtained from the special experiment on the shallow ice-covered lake Vendyurskoe (Karelia, Russia) between 27 March and 6 April 2020. The under-ice convection (most intense during 28–31 March and 4–6 April, when solar radiation was maximal) was clearly observed, with the convectively-mixed layer's (CML) thickness varying from 3 to 6 m. The details of experimental setup are presented in [14].

The measurements were carried out near the northern shore of the lake, the location of the experimental complex is marked by a triangle on Figure 2a; the depth at this location was ~7 m. The instrumental complex included a thermistor chain with 13 temperature sensors (RBR Ltd., accuracy ± 0.002 °C, measurement interval 10 s). The vertical temperature profile is schematically presented in Figure 2b; it clearly demonstrates the splitting of the water body into three sublayers (thin underice gradient sublayer, CML, and the underlying stratified zone), which is typical for developed convection. The CML's lower boundary was determined by thermistor chain data as a depth of the isotherm with a value exceeding the average temperature of the CML by 0.05 °C.

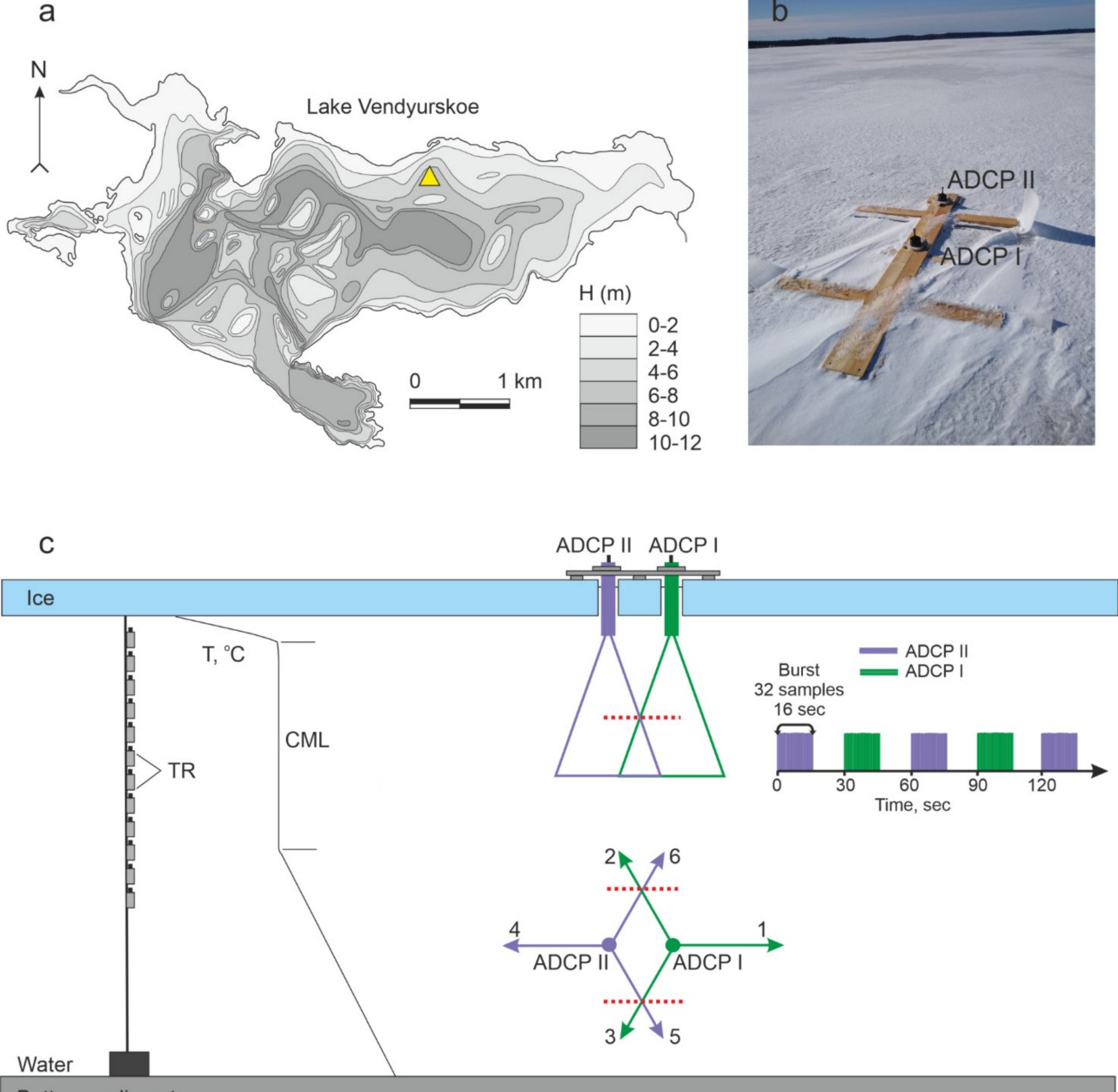

**Figure 2.** (**a**) Bathymetry of the Lake Vendyurskoe with indication of the measuring complex (yellow triangle). (**b**) Two ADCPs anchored on the ice of Lake Vendyurskoe in Spring 2020. (**c**) Schematic vertical distribution of temperature during springtime underice convection and scheme of the measuring complex. Indexes 1, 2, 3—the beams of the first ADCP, and 4, 5, 6—the second. Red dashed lines serve as the markers of beams' intersection points.

The most essential feature is connected with installing two rigidly connected down-looking ADCPs (2 MHz HR Aquadopp current velocity profiler, Nortek AS, Norway). Both devices were installed on the ice (Figure 2b) with emitters located 3 cm below the lower ice boundary (Figure 2c). The X-axes of both devices were aligned with the separation vector between two emitters, but were oppositely directed (Figures 1 and 2c). Due to the choice of this specific configuration one (X-axes are oriented towards each other) or two (X-axes are oriented away from each other) pairs of beams have intersection points. Figure 2c illustrates the second variant, with the beams in question being 3, 5 and 2, 6. The first variant of the devices' settings was realized from 17:00 on 27 March to 9:30 on 30 March, and the second between 10:00, 30 March and 10:00, 6 April 2020. In both cases the depth of the intersection points was the same (~1.6 m). By fitting the distance between the emitters. The presence of intersection points, as was shown in [14], is a key feature for deriving full stress tensor from coupled-ADCP data.

For both variants of coupled-ADCP setup, the signal discreteness was one minute (32 pulses with a frequency of 2 Hz) and the depth scanning range was 2.875 m (115 cells with a size of 25 mm). To exclude the mutual influence of the two ADCPs, the emitters were set in an asynchronous mode with a 30 s delay (Figure 2c). Then the radial velocities were averaged over 16 s active series; further processing was carried out using these averages, for which the same designations $b_i$ were used. The root-mean-square error of $b_i$ values varied in the range (0.1–0.5) mm/s.

Data processing was carried out in accordance to algorithm presented above. The details of the averaging procedure and the choice of averaging interval (100 min.) are presented in [14]. For specificity, only the results, which correspond to the active solar radiation period of 4–6 April and a depth of 1.6 m (corresponding to the position of beams intersection point) are presented below. For the rest of the data (another dates and depths) the results are similar.

The calculated dynamics of beam velocity intensities $\left\langle b_i'^2 \right\rangle$ and interbeam correlations $\left\langle b_i' b_j' \right\rangle$ are presented by Figure 3. The daytime maximums of the intensities reached the values (1–2) mm$^2$/s$^2$, whereas interbeam correlations, remaining statistically significant, varied roughly from one third to one half of their limits.

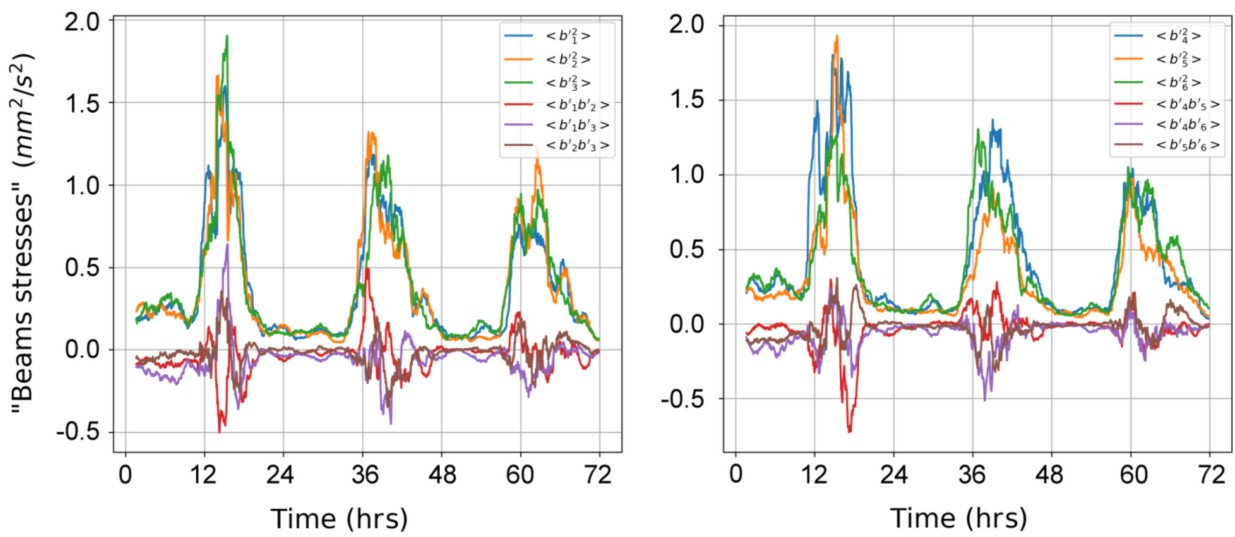

**Figure 3.** The calculated dynamics of beam velocity intensities and interbeam velocity correlations for both devices. Time readings from 4 April, 00:00.

The next step included the calculations of structure functions $D_{LL}$. During daylight time the structure function curves clearly demonstrate the presence of the inertial interval for all six beams. For illustration, the daytime sequence of the calculated SF for 4 April

is presented by Figure 4 (the curves were averaged by all six beams). The upper bound of the inertial interval reached values up to 1 m, which is not much less as compared to the distance between beams at the depth under consideration. This fact gives grounds (see point 3 of the algorithm above) for involving Equation (6) as a missing equation. So Equation (8) may be used for stress estimations.

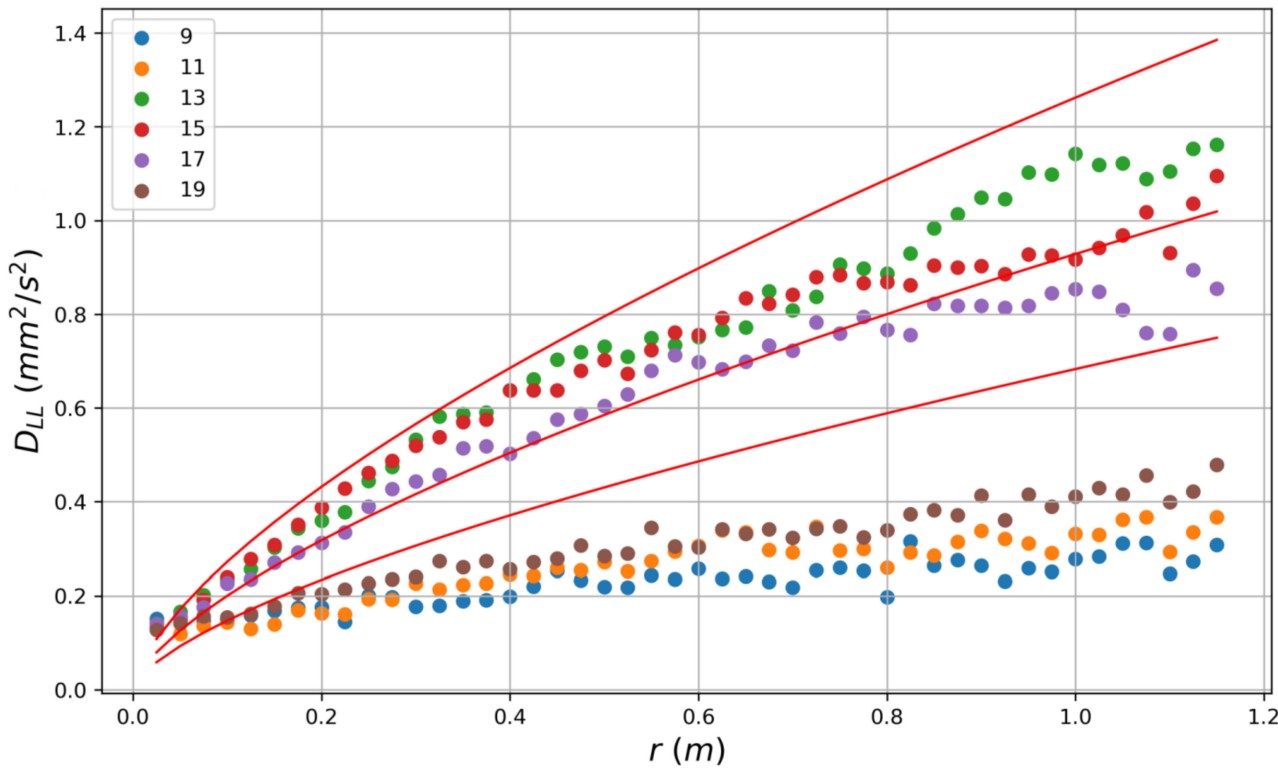

**Figure 4.** The set of longitudinal SF (averaged over six beams) calculated with a two-hour step for the time interval (09:00–18:00) 4 April 2020. Time averaging over 100 min. Solid lines represent the series of Kolmogorov curves $D_{LL} = C \, \varepsilon^{2/3} \, r^{2/3}$ with $\varepsilon$ increasing from $0.2 \cdot 10^{-9}$ to $0.5 \cdot 10^{-9}$. The labels correspond to the measurement time (a.m.).

The calculations of the stresses were carried out by Equation (8) for both devices separately. The results of these independent computations demonstrate qualitative agreement, as Figure 5 illustrates.

Daily maximums of pulsation intensities along axes *X, Y, Z* achieved the values 7, 4 and 1 $mm^2/s^2$ correspondently. The value of the anisotropy coefficient $\langle u_3'^2 \rangle / \langle u'^2 \rangle$ was subjected to irregular oscillations within the range (0.05–0.30).

The estimations of standard deviation for stresses were carried out at the same way as presented in [14]. The errors varied from 15% for $\langle u_1'^2 \rangle$ to 25–30% for off-diagonal stresses. It is also worthy to note another criterion of physical sustainability of the results. This criterion—the so-called realizability condition—includes the positive definiteness of pulsations intensities and the Cauchy–Schwarz inequalities $\left\langle u_i' u_j' \right\rangle^2 \leq \left\langle u_i'^2 \right\rangle \left\langle u_j'^2 \right\rangle$ (here no summation over repeated indices). The fulfillment of both restrictions is the crucial point for low-energetic flow computations. In our case, the violations of this criterion were fixed during time intervals (presented by vertical red lines on the middle image of the top panel, Figure 5), which cover less than 5% of the whole observational period. Most of these intervals belong to nighttime, when the turbulence was sufficiently suppressed.

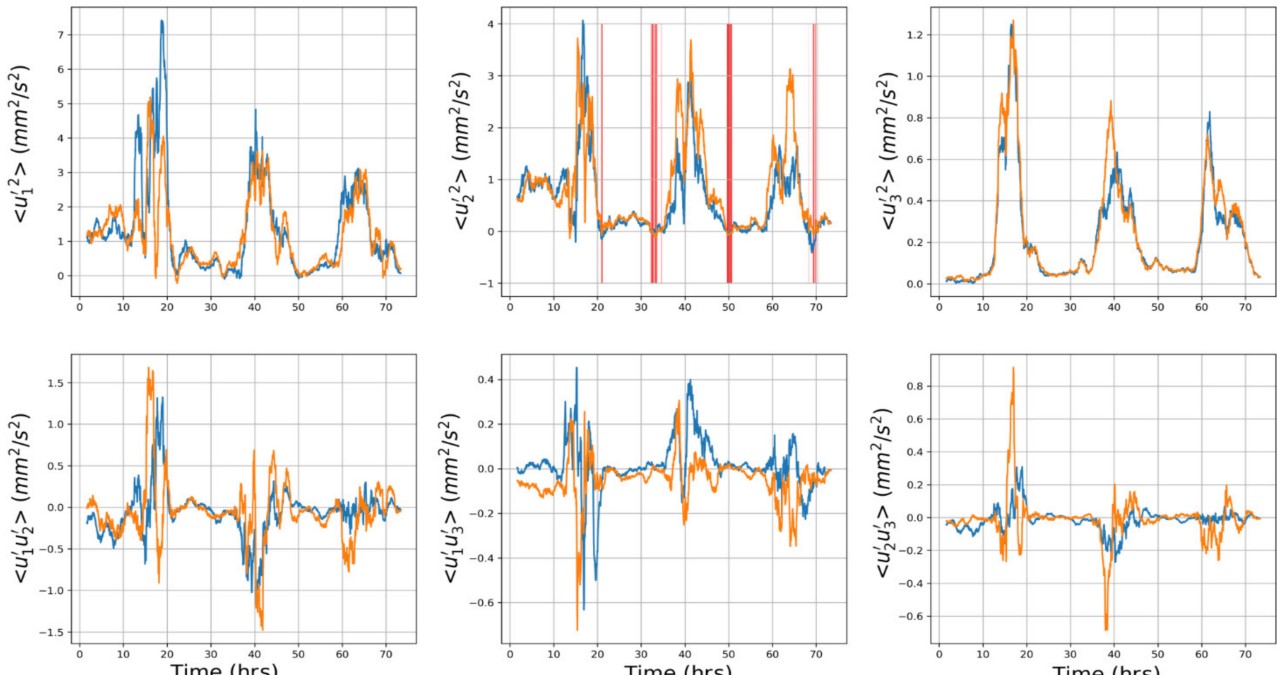

**Figure 5.** The dynamics of turbulent stresses calculated independently for each device. Top panel—pulsation intensities along axes *X*, *Y*, *Z*. Bottom panel—off-diagonal stresses. The time intervals, when realizability conditions were violated, are marked by the vertical red lines on the middle image of top panel.

As was mentioned in the introduction, the same experimental data were used in [14], but the yielding of stresses was carried out by the coupled-ADCPs method. The comparison of these alternative computations may serve as additional verification of the new method presented in this paper. The results of this comparison are presented in Table 1 and Figure 6. The computations by the new method are presented as the average values of the results, obtained for each device separately.

**Table 1.** Comparison of stress computations by two independent methods.

| Stress Component | Correlation Coefficient, $r$ | Coefficient of Determination, $R^2$ | Linear Regression Coefficient |
|---|---|---|---|
| $\langle u_1'^2 \rangle$ | 0.96 | 0.92 | 1.31 |
| $\langle u_2'^2 \rangle$ | 0.98 | 0.96 | 1.03 |
| $\langle u_3'^2 \rangle$ | 0.92 | 0.85 | 0.61 |
| $\langle u_i'^2 \rangle$ | 0.99 | 0.98 | 1.14 |

For all three pulsation intensities (diagonal components of the stresses matrix) the correlation coefficient $r$ is higher than 0.9. At the same time, the new single-ADCP method gives the values 1.31 for $\langle u_1'^2 \rangle$ and 0.61 for $\langle u_3'^2 \rangle$, as compared to the coupled-ADCPs method. The values of the component $\langle u_2'^2 \rangle$, calculated by both methods, are practically identical (with deviations within 3%). The best fitting (Figure 7) was observed for turbulent kinetic energy $\langle u_i'^2 \rangle$ (TKE) for which coefficient of determination $R^2$ achieved the value 0.98 (Table 1).

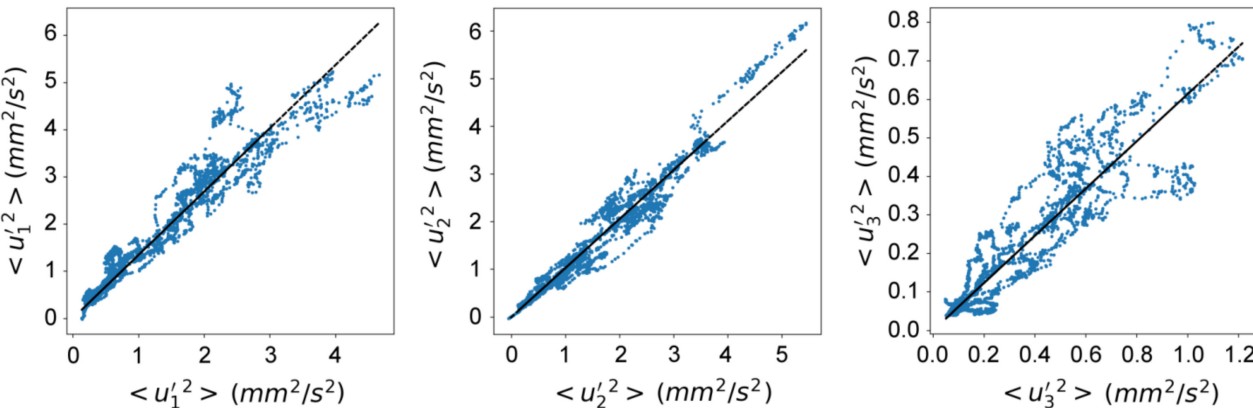

**Figure 6.** Correlations between the pulsation intensities, derived by two independent methods. Projection of each point on the *X* and *Y* axes represent the values, obtained by the coupled-ADCPs and single-ADCP methods correspondently. The linear regression curves are presented by black dashed lines.

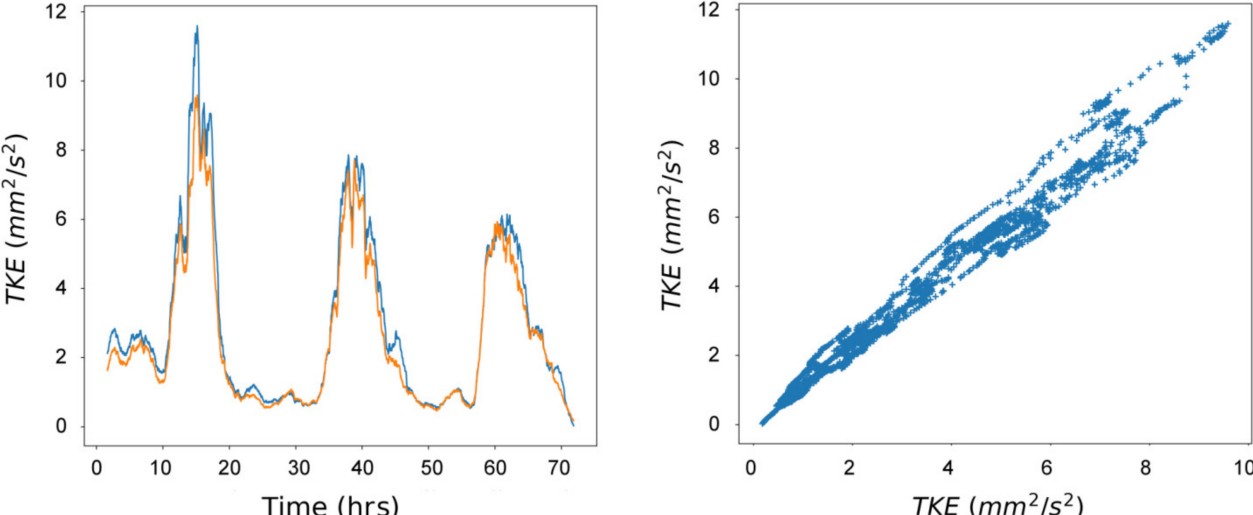

**Figure 7.** Correlations between the TKE values, derived by two independent methods. Projection of each point on the *X* and *Y* axes represent the values, obtained by the coupled-ADCPs and single-ADCP methods respectively.

## 4. Discussion

The method for yielding turbulent stresses presented in this paper is not restricted to the case of three-beam ADCP. Moreover, with other conditions being equal, four- or five-beam devices, presumably, give some preferences for method implementation. First of all, with such devices, more equations of type (2) and (6) become available. The resulting system turns out to be overdetermined, but can be solved in a least-square sense, as was done in [13] for the case of coupled ADCPs. Though such a solution is only an approximation, one may expect that the procedure does not reduce the accuracy of stress estimation, due to the increase of information involved.

The additional advantage of Janus and Janus+ versions is that the angle $2\alpha$ between beams is smaller, as compared to the three-beam device. As a result, the distance between beams for given depth becomes smaller too, so the depth range, where the correlations of the beams velocity increments satisfy the local isotropy and homogeneity requirements and Equation (6) are valid, becomes wider.

**Author Contributions:** Conceptualization, S.B.; methodology, S.B. and R.Z.; software, S.B.; validation, S.B. and N.P.; formal analysis, S.B. and G.Z.; investigation, R.Z., G.Z. and N.P.; resources, G.Z.; data curation, S.B.; writing—original draft preparation, S.B. and G.Z.; writing—review and editing, S.B. and G.Z.; visualization, S.B. and G.Z, supervision, S.B.; project administration, G.Z.; funding acquisition, G.Z. All authors have read and agreed to the published version of the manuscript.

**Funding:** This research was funded by the Russian Science Foundation project No 21-17-00262 "Mixing in boreal lakes: mechanisms and its efficiency".

**Data Availability Statement:** All data created or used during this study are available by request to authors.

**Acknowledgments:** Authors are thankful to G. Kirillin for fruitful discussions.

**Conflicts of Interest:** The authors declare no conflict of interest.

## Appendix A. Derivation of the Relationships between Structure Functions

Consider the plane formed by beams 1 and 2 and introduce the rectangular coordinate system as indicated in Figure A1. In this frame of reference, the beams velocities are presented by:

$$\begin{cases} b_1 = -u_x sin\alpha + u_z cos\alpha \\ b_2 = u_x sin\alpha + u_z cos\alpha \end{cases}$$

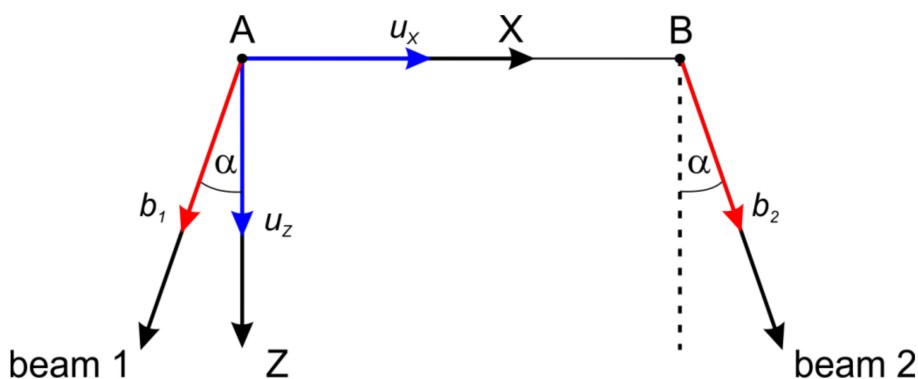

**Figure A1.** The reference frame for the plane including beams 1 and 2.

Substituting these expressions in (3), one obtains:

$$\tilde{D}_{12} = \langle ((u_z(A) - u_z(B))cos\alpha - (u_x(A) - u_x(B))sin\alpha)((u_z(A) - u_z(B))cos\alpha + (u_x(A) - u_x(B))sin\alpha)\rangle,$$

or, equivalently:

$$\tilde{D}_{12} = -\Big\langle (u_x(A) - u_x(B))^2 \Big\rangle sin^2\alpha + \Big\langle (u_z(A) - u_z(B))^2 \Big\rangle cos^2\alpha - < (u_x(A) - u_x(B)) \\ (u_z(A) - u_z(B))sin\ \alpha\ cos\alpha + (u_z(A) - u_z(B))(u_x(A) - u_x(B))sin\ \alpha\ cos\alpha \tag{A1}$$

The first term in (A1) includes the so-called longitudinal SF $D_{LL} = < (u_x(A) - u_x(B)^2 >$, which is associated with the increments of the velocity components aligned with separation vector $\vec{r} \equiv \vec{AB}$ between points A and B. The second term, in turn, may be presented through the transverse SF $D_{NN} = \Big\langle (u_z(A) - u_z(B)^2 \Big\rangle$, which is defined through the increments of orthogonal velocity components.

The SF of general type is defined as $D_{ij} = \Big\langle (u_i(A) - u_i(B))(u_j(A) - u_j(B)) \Big\rangle$. Here velocity components $u_x$, $u_y$, $u_z$ are numbered from 1 to 3. For locally isotropic and homogeneous turbulence, $D_{ij}$ is presented through $D_{LL}$ and $D_{NN}$ by the expression [16]:

$$D_{ij}\left(\vec{r}\right) = (D_{LL}(r) - D_{NN}(r))\frac{r_i r_j}{r^2} + D_{NN}(r)\,\delta_{ij} \tag{A2}$$

The last two terms in (A1) include the cross-correlations of the aligned and orthogonal velocity components, and so are proportional to $D_{13}$. Due to (A2) both these terms turns to 0, if one takes into account the presentation $(r, 0, 0)$ for vector $\vec{r}$.

Finally the relation (A1) is transformed to:

$$\widetilde{D}_{12} = D_{NN}cos^2\alpha - D_{LL}sin^2\alpha$$

For locally isotropic and homogeneous turbulence $D_{NN} = 4\,D_{LL}/3$, so one obtains the following presentation of $\widetilde{D}_{12}$ through the longitudinal SF: $\widetilde{D}_{12} = \left(\frac{4}{3}cos^2\alpha - sin^2\alpha\right)D_{LL}$.

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
