# Peer review of "Deriving Six Components of Reynolds Stress Tensor from Single-ADCP Data"

_water, doi:10.3390/w13172389_

Round 1

Reviewer 1 Report

See attached PDF

Author Response

Reply to Reviewer 1.

The article “Deriving six components of Reynolds stress tensor from a single ADCP data”.

Sergey Bogdanov, Roman Zdorovennov, Nikolay Palshin, Galina Zdorovennova.

We are thankful to the Reviewer for the careful reading of the manuscript and valuable comments and advices.

Point 1:

Please introduce the pro and cons of the traditional methods and the proposed method in calculating Reynolds stress.

Response 1:

We stressed the advantage of the new method in lines 66-68.

Point 2:

Line 148. The step-by-step calculation is confusing. Some better explanations could be helpful.

Response 2:

It seems, that the derivation of  (eq. (5), ij) is the most challenging. So, we rearranged the text, devoted to structure functions. The Appendix was expanded by including the proper definitions and relationships.

Point 3:

The validation can be further improved using some error analysis or uncertainty discussion.

Response 3:

We modified the figure 5 (former Fig. 4) by including the additional panel, which present the results of the off-diagonal stresses. Additionally, we presented on this figure those time intervals, which correspond to realizability conditions violations.

Reviewer 2 Report

The author presents a good alternative method for Reynolds stresses using previously neglected ADCP data.  The method is well presented. This reviewer is fully supporting for acceptance.

Some comments:

  1. Please introduce the pro and cons of the traditional methods and the proposed method in calculating Reynolds stress.
  2. Line 148. The step-by-step calculation is confusing. Some better explanations could be helpful.
  3. The validation can be further improved using some error analysis or uncertainty discussion.

Author Response

We are thankful to the Reviewer for the careful reading of the manuscript and valuable comments and advices. Our responces are presented in the attached file

Round 2

Reviewer 2 Report

This reviewer is satisfied with all the changes and responses.